# Expression of Toll-like Receptors on the Immune Cells in Patients with Common Variable Immune Deficiency after Different Schemes of Influenza Vaccination

**DOI:** 10.3390/v15102091

**Published:** 2023-10-14

**Authors:** Aristitsa Mikhailovna Kostinova, Elena Alexandrovna Latysheva, Nelly Kimovna Akhmatova, Anna Egorovna Vlasenko, Svetlana Anatolyevna Skhodova, Ekaterina Alexandrovna Khromova, Andrey Viktorovich Linok, Arseniy Alexandrovich Poddubikov, Tatyana Vasilievna Latysheva, Mikhail Petrovich Kostinov

**Affiliations:** 1Federal State Autonomous Educational Institution of Higher Education I.M. Sechenov First Moscow State Medical University of the Ministry of Health of the Russian Federation (Sechenov University), Trubetskaya Str. 8/2, 119991 Moscow, Russiaarseniypoddibikov@gmail.com (A.A.P.); monolit.96@mail.ru (M.P.K.); 2National Research Center—Institute of Immunology Federal Medical-Biological Agency of Russia, Kashirskoe Shosse, 24, 115478 Moscow, Russiatvlat@mail.ru (T.V.L.); 3Pirogov Russian National Research Medical University, Ostrovitianov Str. 1, 117997 Moscow, Russia; 4Russian Federal State Budgetary Scientific Institution «I.I. Mechnikov Research Institute of Vaccines and Sera», Malyi Kazenniy Pereulok, 5a, 105064 Moscow, Russiakate.khromova@mail.ru (E.A.K.); 5Federal State Budgetary Educational Institution of Higher Education “Samara State Medical University” of the Ministry of Healthcare of the Russian Federation, Chapaevskaya Street, 89, 443099 Samara, Russia

**Keywords:** quadrivalent adjuvanted influenza vaccines, toll-like receptors, influenza, CVID, azoximer bromide

## Abstract

Background: for the first time, the effect of one and two doses of adjuvanted influenza vaccines on toll-like receptors (TLRs) in patients with common variable immunodeficiency (CVID) was studied and compared (primary vaccination with one vs. two doses, primary vs. repeated vaccination). Materials and methods: Six patients received one dose of quadrivalent adjuvanted influenza vaccine during the 2018–2019 and 2019–2020 influenza seasons, and nine patients with CVID received two doses of trivalent inactivated influenza vaccine during 2019–2020. Expression of TLRs was measured by flow cytometry. Results: The expression of toll-like receptors in patients with CVID was noted both with repeated (annual) administration of the influenza vaccine and in most cases was accompanied by an increase in the proportion of granulocytes (TLR3 and TLR9), lymphocytes (TLR3 and TLR8), and monocytes (TLR3 and TLR9). When carried out for the first time as a simultaneous vaccination with two doses it was accompanied by an increase in the proportion of granulocytes, lymphocytes expressing TLR9, and on monocytes—TLR3 and TLR9. Conclusion: in CVID patients, the use of adjuvanted vaccines is promising, and research on the influence of the innate immunity and more effective regimens should be continued.

## 1. Introduction

Inborn errors of immunity (IEI; primary immunodeficiencies (PIDs)) encompass a heterogeneous group of orphan diseases (prevalence 1 to 5 in 1000), which are based on approximately 500 currently known genetic defects of the immune system [1]. The most common clinical manifestation of IEI is an increased risk of recurrent and potentially life-threatening infections due to defects of immunological defense mechanisms and its critical pathway. Immunodeficiencies affecting humoral immunity (antibody synthesis) engage more than 50% of all diseases in the structure of IEI. The main treatment for patients with impaired antibody synthesis is lifelong replacement therapy with intravenous immunoglobulin (IVIG), containing a wide variety of donor class G immunoglobulins that reduce susceptibility to infectious agents. However, this treatment is not expected to protect recipients from all currently circulating infections as antibodies to rare or highly variable infectious agents are often absent in donor plasma [2].

Nowadays many patients with IEI survive into adulthood, and regular and effective therapy allows them to have a high quality of life that does not differ from healthy people: they work and study, have hobbies, make friends, and have families. But at the same time, this active lifestyle in the community leads to frequent contact with respiratory viruses, and the high susceptibility of IEI patients contributes to exacerbations of chronic sinopulmonary diseases. That is why we can say that the prevention of infections is particularly acute.

Vaccination is recognized in the whole world as the most effective means of protection against infectious diseases. Most countries have already implemented vaccination programs for the population against respiratory infections that help to reduce the risk of contamination and morbidity, both through the direct protection of each vaccinated individual and through the development of herd immunity [3]. However, vaccination coverage among immunocompromised people still remains below the WHO recommendations, even in developed countries [4]. This is facilitated by many socially significant factors, such as the influence of the media, existing misconceptions about the effectiveness and safety of vaccination, low awareness of medical workers, and the doctor–patient relationship [5]. Additionally, certain features of the pathogens themselves, for example, high mutational variability, contributes to the risk of pandemics even nowadays.

Over the past 100 years, four influenza pandemics have occurred on the planet, with the 1918 pandemic, caused by the influenza A/H1N1 virus, being the most devastating as it claimed the lives of more than 40 million people [6]. The last influenza pandemic “engulfed” the world in 2009–2010. Therefore, vaccination against the influenza virus still remains actual and relevant.

Immunization is the only way to form protection against seasonal influenza, not only in healthy people but also in patients with IEI. In order to reduce morbidity and mortality from vaccine-preventable infections in immunocompromised patients, for the first time in 2013 in the guidelines of the American Society of Infectious Diseases and in the world in 2015, patients with IEI were recommended to undergo annual vaccination against the influenza virus regardless of the type of the immune defect [7,8]. However, currently available data on the formation of post-vaccination immunity in this cohort of patients are limited and in some cases contradictory. It is not only due to different vaccination schemes, the use of different vaccines, but also because of the various examined parameters of the immune system. Considering that in patients with IEI there is a need to increase the effectiveness of vaccines, adjuvant vaccines which enhance a specific immune response are preferable [9,10].

However, no study has been conducted comparing both immunogenicity of different influenza vaccination regimens in patients with impaired humoral immunity and the clinical significance of immunization on the course of the underlying disease of patients for the purposes of practical health care, and not only from the standpoint of scientific research. Approximately 10–20% of patients from the most common group of patients with a defect in the humoral immunity—common variable immune deficiency (CVID)—have a residual response to vaccination against protein antigens and, to a lesser extent, against polysaccharide antigens. However, post-vaccination immunity in patients with CVID has not been sufficiently studied, which is due to the different use of various vaccine preparations, immunization regimens, determined markers, research methods, and the lack of clear standards for many indicators of innate immunity. Moreover, the effect of quadrivalent adjuvant influenza vaccines on the parameters of innate and adaptive immunity has not been studied.

Innate immunity is the first line of the immune system defense, both in evolutionary terms and in terms of time response, which develops in the first hours and days after exposure to a pathogen. The speed of the response is realized through the involvement of already existing stereotypical mechanisms for recognizing specific molecular structures of pathogens. It is the recognition of molecular structures of microbial origin that is the key component of the immune response that initiates inflammation [11]. This response is mediated by a special family of receptors that recognize the most common molecular patterns (PAMPs—pathogen associated molecular patterns) of microorganisms (viruses, bacteria, parasites, etc.) and are called PRRs (pattern recognition receptors) [12]. PRRs are genetically stable. Functionally, they can be divided into two groups: endocytic and signaling. Endocytic PRRs (mannose receptors and scavenger receptors) have been known in immunology for a long time and provide phagocytosis processes with subsequent delivery of the pathogen inside the phagosome to lysosomes, providing the start of an adaptive immune response. Among signaling PRRs are three families of great importance: toll-like (TLR), nod-like (NLR), and rig-like receptors (RLR). TLRs were the first PRRs to be identified, and that is why they are the most studied.

Fifteen TLRs have been described in mammals and humans. They are located on the membrane, in endosomes, or in the cytoplasm of cells that serve as the first line of defense (neutrophils, macrophages, dendritic, endothelial, and epithelial cells of the skin and mucous membranes) from pathogens.

TLR1, TLR2, TLR4, TLR5, and TLR6 are expressed on the extracellular surface and recognize microbial wall components including lipopolysaccharides, lipopeptides, and flagellin. TLR3, TLR7, TLR8, and TLR9 are mostly located in endosomal compartments; this arrangement allows these receptors to recognize the DNA and RNA breakdown products of viral and bacterial origin. TLRs are expressed in a variety of cell types, including monocytes, phagocytes, dendritic cells, and B cell subpopulations. TLR1, TLR2, TLR6, TLR7, TLR9, and TLR10 are expressed on human B cells.

While B cells are traditionally considered to play a key role in adaptive immunity due to their ability to produce antibodies, activation of innate immune receptors also expressed on B cells provides a co-stimulatory effect that promotes both their function and survival [13] and the coordination of innate and adaptive immune signals, resulting in a wide range of cellular responses. Memory B cells are usually formed in germinal centers in response to T-dependent or T-independent antigens. Like other antigen-presenting cells, B cells express various TLRs [14,15]—conservative membrane proteins which provide alternative ways to activate B cells [16].

One of the most powerful stimulators of B cell activation and maturation are endosomal TLRs in connection with single-stranded RNAs or various synthetic agonists (TLR7) and unmethylated CpG motifs in microbial DNA (CpG DNA) (TLR9) [17]. Binding of TLR9 by CpG DNA has been shown to activate normal B cells, increase the expression of costimulatory molecules, trigger the secretion of IL-6 and IL-10, and mediate T independent isotype switching and production of specific antibodies independent of B cell receptor (BCR) binding [18,19,20,21,22]. Cross-linking of the B cell receptor (BCR) leads to rapid activation of their expression. Naive B cells express low levels of TLR while memory B cells constitutively express TLR7, TLR8, and TLR9 at higher levels [23,24,25,26]. These differences between naive and B memory cells are connected with different adaptive functions: memory B cells express higher levels of TLR and have a more ability to differentiate into plasma cells via TLR stimulation compared to naive B cells [27]. On the contrary, it is suggested that prolonged PAMP activation has undesirable consequences for the organism since the TLR signaling pathway is controlled by a variety of feedback mechanisms [28].

Both in vivo and in vitro experiments have shown that switching of B cells to IgG isotypes requires the simultaneous presence of at least two signals along with BCR involvement: TLR activation and the involvement of CD40 and/or IFN-alpha [29]. These observations have led to studies suggesting that TLR activation may provide a long-lasting stimulus important for maintaining memory B cell proliferation and differentiation into mature antibody-secreting cells, which are initially induced by BCR and by T cells [30,31]. Binding and activation of TLR7 and TLR9 can serve as a signal for the start of B cell differentiation after antigen stimulation via BCR. However, there are data reporting defects in TLR7 and TLR9 on B cells from patients with CVID [17]. Since TLR activation seems to be an essential mechanism for the activation and subsequent survival of memory B cells [32], a study of TLR9 defects in patients with CVID revealed that their B cells were not activated by the CpG ODN ligand even upon costimulation of BCR, as well as by secretion of IL-6 and IL-10; thus, there was no TLR activation, low proliferation of B cells, absence of their maturation, isotype switching, and production of IgG and IgA [17].

Thus, the exact role of TLRs in B cell biology is not clear: are they necessary for the development of some normal humoral immunity pathways, or does stimulation of TLRs serve more as an adjuvant to existing functions? It is believed that TLR signaling pathways may provide secondary stimuli for B cell development, but other molecular mechanisms may compensate for defective signaling through these innate receptors [33]. A more detailed understanding of the innate immune status upon ligand-activated TLRs is needed to identify specific defects in innate immune responses in CVID patients that may explain the likely variability in clinical symptoms.

## 2. Materials and Methods

In an open-label, single-center, non-randomized, prospective, cohort, controlled study the effect of influenza tetravalent inactivated subunit adjuvanted vaccine on the expression of endosomal toll-like receptors on immunocompetent cells in healthy volunteers and patients with CVID using different influenza vaccination schemes were investigated.

### 2.1. Participants

In 2018, from the registry of the Institute of Immunology that includes all adult patients with IEI from the whole Russian Federation, 297 outpatient records of patients with IEI were analyzed, and 203 patients were selected with a diagnosis of CVID, which was established in accordance with the criteria of the European Society on Immunodeficiency Disorders (ESID). However, due to the comorbidity of patients and in accordance with strict criteria for inclusion in the study, as well as the need to relieve exacerbations when patients were hospitalized, the annual change in the strain composition of influenza vaccines, and the limited timing of influenza vaccination (autumn–winter period), immunization was administered to only 15 patients with CVID.

### 2.2. Inclusion Criteria

Confirmed diagnosis of CVID in accordance with diagnostic criteria established by the European Society for Immunodeficiency Diseases and the American Academy of Allergy, Asthma and Immunology for the diagnosis and treatment of IEI.IVIG therapy no later than 28 days before vaccination and no earlier than 21 days after it; that is, there was a break between two subsequent administrations of immunoglobulins of at least 7 weeks.Signed informed consent.Excluded causes of secondary hypogammaglobulinemia.No symptoms of flu or flu-like illness in the past six months.No glucocorticosteroid or other immunosuppressive therapy admission at the time of the study and 3 months before the start.The absence of symptoms of protein-losing enteropathy as well as any suspicion of oncological or lymphoproliferative disease in patients with CVID at the time of the study.No level of specific antiviral antibodies in protective titers (>1:40) in pre-vaccination blood samples.Individuals with cognitive or behavioral impairments, psychiatric disorders, or alcohol abusers that could preclude participation in the study were excluded.Vaccination against any other infections within 1.5–2 months prior to enrollment in the study.All contraindications for use in accordance with the instructions (a history of allergy to egg whites or any component of the study vaccine, symptoms of acute infection at the time of vaccination, pregnancy, etc.) were followed.

### 2.3. Vaccination Schemes

Thus, in the 2018–2019 influenza season at the Department of Immunopathology in the National Research Center—Institute of Immunology Federal Medical-Biological Agency of Russia and in the laboratory of the Mechanisms of Immune Regulation in Mechnikov Research Institute of Vaccines and Sera of the Ministry of Health of the Russian Federation in Moscow, 6 patients with a diagnosis of CVID were involved in the study. They received a single dose (0.5 mL) of a quadrivalent subunit adjuvant vaccine (aQIV) against the influenza virus “Grippol Quadrivalent” (NPO Petrovax Pharm LLC, Moscow, Russia). Exactly one year later 5 out of these 6 patients were re-vaccinated with the trivalent subunit adjuvant influenza virus vaccine Grippol Plus (NPO Petrovax Pharm LLC, Russia) from the Grippol family of vaccines. The mean age of patients with CVID was 36.6 ± 2.03 years. In the next 2019–2020 influenza season 9 more patients with CVID were immunized simultaneously with a double dose (2 × 0.5 mL) of the trivalent subunit adjuvant influenza virus vaccine (aTIV) Grippol Plus.

Thirty-two healthy volunteers aged 19 to 48 years (32.56 ± 1.67 years) in the 2018–2019 season were also vaccinated with a single dose of aQIV according to the same schemes as patients with CVID and served as a control group to determine the working concentrations for immunological parameters. All participants had not been vaccinated against influenza in the previous two seasons (2016–2017, 2017–2018), nor were there any confirmed cases of influenza reported.

### 2.4. Used Vaccines

Grippol Quadrivalent (NPO Petrovax Pharm LLC, Moscow, Russia) and Grippol Plus (NPO Petrovax Pharm LLC, Moscow, Russia) vaccines were available for immunization of people aged 18 to 60 years during the 2018–2019 and 2019–2020 flu season, respectively, and included strains that met the 2018–2019 WHO recommendations for the northern hemisphere for quadrivalent vaccines (A/Michigan/45/2015 (H1N1)pdm09-like virus; A/Singapore/INFIMH-16-0019/2016 (H3N2)-like virus; B/Colorado/06/2017-like virus (B/Victoria/2/87lineage); B/Phuket/3073/2013-like virus (B/Yamagata/16/88 lineage)), and in 2019–2020 for trivalent vaccines (A/Brisbane/02/2018 (H1N1)pdm09-similar virus; A/Kansas/14/2017 (H3N2)-like virus; B/Colorado/06/2017-like virus (lineage B/Victoria/2/87)). The vaccines contained 5 μg of hemagglutinin of each included influenza strain and 500 μg of azoximer bromide (without preservative). The key benefit of the vaccine is a three-fold decreased amount of hemagglutinin protein as compared to traditional technologies due to the use of Polyoxidonium (azoximer bromide)—a water-soluble, high-molecular immune system adjuvant that enhances the immune response to vaccination.

### 2.5. Methods

The expression of TLRs on granulocytes (CD45+CD15+), lymphocytes (CD45+CD19+), and monocytes (CD45+CD14+) in the peripheral blood in patients with CVID and healthy patients before vaccination and 24 ± 3 days after was studied in vitro by flow cytometry using mAbs to TLR3-PE, TLR8-FITC, and TLR9-PE (e-Bioscience, San Diego, CA, USA) on the Cytomix FC 500 flow cytometer (Beckman Coulter, Indianapolis, IN, USA) according to the method described in the manufacturer’s instructions.

The study was conducted according to the Russian Federation National Standard Protocol ГOCTP 52379-2005 Good Clinical Practice and International GCP standards [34]. The study was based on the ethical principles and recommendations of the WHO and the Russian Ministry of Health.

### 2.6. Statistics

Checking the normality of the distribution of signs was carried out by the Shapiro–Wilk normality test. As a result, significant deviations from the normal distribution were revealed in the distribution of signs. Analysis of the dynamics of signs and comparison between study groups was carried out using the construction of a robust linear mixed effects model (RLMEM) [35]. The statistical significance of the model coefficients was determined using Satterthwaite’s degrees of freedom approximation [36]. A posteriori comparisons (between groups at control points and between control points for each group) were carried out by constructing the corresponding contrasts based on the calculated model using the emmeans package [37].

When comparing the formation of post-vaccination immunity after the primary vaccination with one and two doses, the time after the introduction of the vaccine preparation (24 ± 3 days) and one or two doses of the vaccine were determined as fixed factors, individual patients were set as accidental factors. When comparing the formation of post-vaccination immunity after primary and after re-vaccination with a single dose of the drug, the time after the introduction of the vaccine drug (24 ± 3 days), and the stage of vaccination (primary/revaccination), individual patients were determined as fixed factors (the associated nature of the samples was taken into account both in time and stages of vaccination) were given as random factors. Comparison of delta cell number changes between study groups was performed using the Mann–Whitney test for unrelated samples (one and two doses of vaccine) and using the paired Wilcoxon test for related samples (primary and revaccination with one dose).

Differences were considered statistically significant at *p* ≤ 0.05 and insignificant at *p* ≥ 0.10; in intermediate cases (0.05 < *p* < 0.1), trends towards differences were discussed [38]. Calculations and graphical constructions were made using the GraphPad Prism program (v.9.3.0 licensed GPS-1963924) and the statistical environment R (v.3.6, GNU GPL2 license).

## 3. Results

No one in the control nor the CVID groups came down with influenza virus infection within the 24 days post-vaccination in the 2018–2019 and 2019–2020 seasons.

### 3.1. Comparing the Content of TLR3, TLR8, and TLR9 in Patients with CVID after the Primary Vaccination and a Year Later with One Dose of the Quadrivalent Vaccine

#### 3.1.1. Granulocytes

The proportion of granulocytes expressing TLR3 and TLR9 after the primary vaccination decreased statistically significantly—from 93.8 (91.3–95.0)% to 89.5 (81.9–91.1)% (*p* = 0.03) and from 93.6 (92.4–96.9)% to 86.0 (81.0–91.4)% (*p* = 0.001), respectively. A year after the primary vaccination, these indicators remained statistically significantly lower than the initial values: 87.0 (69.4–89.0)% for TLR3 (*p* < 0.001) and 87.1 (80.0–88.3)% for TLR9 (*p* = 0.001). After the second vaccination one year after the primary, there was a statistically significant increase in the percentage of granulocytes expressing TLR3 and TLR9, in relation to the values before the second vaccination—up to 92.2 (84.4–93.9)% for TLR3 (*p* = 0.02) and 94.2 (86.5–98.0)% for TLR9 (*p* = 0.001). Thus, as a result of the primary vaccination and repeated vaccination one year after, the proportion of granulocytes expressing TLR3 (Figure 1a) and TLR9 (Figure 1b) did not change statistically significantly relative to the initial level.

However, it is worth noting the changes in these indicators after primary and second vaccination: a decrease after the primary vaccination (5.6 [17.8; 0.1]% for TLR3 and 8.8 [12.9; 3]% for TLR9) and growth after the second vaccination (+6.4 [6.7; +24]% for TLR3 and +8.3 [5.9; +21.1]% for TLR9), *p* = 0.04 and *p* = 0.02, respectively), compared with changes after the primary vaccination (Figure 2a,b).

The percentage of granulocytes expressing TLR8 (data not shown) did not show statistically significant changes during the primary vaccination and repeated vaccination one year after and was 93.7 (85.2–96.0)% initially and 89.4 (88.2–93.0)% 24 ± 3 days after repeated vaccination.

#### 3.1.2. Lymphocytes

The percentage of B-lymphocytes expressing TLR3 and TLR8 shows similar dynamics during immunization: no changes after the primary vaccination (*p* = 0.51 for TLR3 and *p* = 0.41 for TLR8) and an increase after the second vaccination (Figure 3a,b). The proportion of lymphocytes expressing TLR3 increased by 2.7 [1.04; +12.7]% in relation to the level before the second vaccination (from 0.60 (0.25–1.85)%, *p* < 0.001) and by 2.6 [0; +13.9]% in relation to the initial level (from 0.25 (0.09–1.73)%, *p* < 0.001). The proportion of lymphocytes expressing TLR8 increased by 1.8 [9.6; +10.2]% comparatively to the level before the second vaccination (from 0.48 (0.30−5.66)%, *p* = 0.08) and by 1.9 [1.4; +8.6]% comparatively to the initial level (from 0.25 (0.10−1.85)%, *p* = 0.02).

Thus, changes in the proportion of lymphocytes expressing TLR3 and TLR8 after primary and after the second vaccination differ (*p* = 0.05 for TLR3 and *p* = 0.08 for TLR8): there is no change after the primary vaccination and an increase after the second vaccination, one year later (Figure 4a,b).

The percentage of lymphocytes expressing TLR9 (data not shown) shows slightly different dynamics—no changes after the primary vaccination, but a statistically significant increase after a year (by +5.9 [+0.8; +16.4]% to 5.70 (0.70–11.45)%, *p* = 0.01 compared to initial) (Figure 3c). After the second vaccination one year later, the proportion of lymphocytes expressing TLR9 did not change statistically significantly (*p* = 0.15 relative to the level before revaccination) and remained higher than the initial level (*p* < 0.001).

#### 3.1.3. Monocytes

The proportion of monocytes expressing TLR3 and TLR9 after the primary vaccination decreased statistically significantly: by 25.1 [71.3; +0.4]% from 53.1 (22.8–75.6)% to 7.3 (2.7–44.0)% for TLR3 (*p* = 0.01) and by 23.2 [66.7; 5.5]% from 66.1 (35.3–80.8)% to 6.3 (1.6–73.0)% for TLR9 (*p* < 0.001). One year later, the proportion of monocytes expressing TLR3 and TLR9 increased slightly (up to 22.1 (5.1–35.7)% for TLR3 and up to 32.9 (22.6–44.1)% for TLR9) but still remained below the initial levels (*p* = 0.05 and *p* = 0.04, respectively). A statistically significant increase was observed after the second vaccination (relative to the level before the second vaccination): by +20.9 [1.6; +62.4]% to 45.4 (28.7 62.8)% (*p* = 0.04) for TLR3 and by +12.6 [8.6; +30.2]% to 50.2 (33.2 64.6)% for TLR9 (*p* = 0.05). As a result of the increase after the second vaccination, the percentage of monocytes expressing TLR3 and TLR9 became comparable to the initial level (*p* = 0.89 for TLR3 and *p* = 0.40 for TLR9) (Figure 5a,b).

The proportion of monocytes expressing TLR8 also decreased 24 ± 3 days after the primary vaccination: by 34.9 [78.7; +1.6]% from 67.5 (13.9–87.4)% to 7.4 (4.6–24.7)%, *p* < 0.001. But a year after the primary vaccination (before the second vaccination), there was an increase to 41.9 (25.4–58.2)% (*p* = 0.07 relative to the level after the primary vaccination). A total of 24 ± 3 days after the second vaccination, the proportion of monocytes expressing TLT9 was 55.6 (42.3–60.9)%; there were no significant changes compared to the level before the second vaccination (*p* = 0.82); it remained comparable to the initial level (*p* = 0.82) and was statistically significantly higher than the value after the primary vaccination (*p* = 0.02) (Figure 5c).

The percentage of lymphocytes expressing TLR9 (data not shown) shows slightly different dynamics—no changes after the primary vaccination, but a statistically significant increase after a year (by +5.9 [+0.8; +16.4]% to 5.70 (0.70–11.45)%, *p* = 0.01 compared to initial) (Figure 3c). After the second vaccination one year later, the proportion of lymphocytes expressing TLR9 did not change statistically significantly (*p* = 0.15 relative to the level before revaccination) and remained higher than the initial level (*p* < 0.001).

It is worth noting the statistically significant multidirectional nature of changes after the primary and after the second vaccination for all examined TLRs on monocytes (*p* = 0.02 for TLR3, *p* = 0.01 for TLR8, and *p* = 0.006 for TLR9): a decrease after the primary vaccination and increase (or no change, as in the case of TLR8) after the second vaccination (Figure 6a–c).

### 3.2. Comparing the Content of TLR3, TLR8, and TLR9 in Patients with CVID after Administration of One Dose and Simultaneously Two Doses of the Adjuvanted Influenza Vaccine

#### 3.2.1. Granulocytes

After vaccination with one dose, there was a statistically significant decrease in the proportion of granulocytes expressing TLR9, from 93.6 (92.4–96.9)% to 86.0 (81.0–91.4)% (*p* = 0.05) and at a borderline level of significance expressing TLR 3, from 93.8 (91.3–95.0)% to 89.5 (81.9–91.1)% (*p* = 0.08) (Figure 7a,b). While after vaccination with two doses, the proportion of granulocytes expressing TLR9 was statistically significantly higher than after vaccination with one dose—96.7 (90.4–98.1)% vs. 86.0 (81. 0–91.4)% (*p* = 0.02). The delta change in the proportion of granulocytes (relative to the initial level) expressing TLR3 and TLR9 was 5.6 [9.8; 1.0]% and 8.8 [12.2; 3.2]% with one dose of the vaccine and +3.5 [2.9; +19.4]% and +4.5 [0.7; +26.5]% after vaccination with two doses (*p* = 0.11 for TLR3 and *p* = 0.03 for TLR9 compared with one dose) (Figure 8a,b).

The percentage of granulocytes expressing TLR8 (data not shown) did not change significantly before or after vaccination and did not depend on the number of doses; it showed comparable dynamics for one and two doses. The medians were (before and after vaccination, respectively) 93.7 (85.2–96.0)% and 88.9 (85.7–91.0)% after one dose, 94.5 (82.7–99.1)% and 89.8 (84.8–97.9)% after two doses of the vaccine.

#### 3.2.2. Lymphocytes

After vaccination with two doses, there was a statistically significant increase in the proportion of lymphocytes expressing TLR9, from 2.70 (0.40−6.60)% to 15.40 (1.90−26.75)% (*p* = 0.02). No such changes were observed after vaccination with one dose. A total of 24 ± 3 days after vaccination with two doses, the proportion of lymphocytes expressing TLR9 was higher than after vaccination with one dose: 15.40 (1.90–26.75)% vs. 0.30 (0.05–1.28)% (*p* < 0.001) (Figure 9).

The change in the proportion of lymphocytes expressing TLR9 as a result of vaccination with two doses was +11 [+1.5; +29.4]%, and as a result of vaccination with one dose it was 0.1 [0.3; +0.3]%, in which the differences were statistically significant (*p* = 0.05) (Figure 10).

The percentage of lymphocytes expressing TLR3 and TLR8 did not change statistically significantly during vaccination, was independent of dose, and showed comparable dynamics for one and two doses.

#### 3.2.3. Monocytes

As a result of vaccination with two doses, the initial level of monocytes expressing TLR3 and TLR9 increased by +10.7 [2.7; +29.2]% from 16.2 (9.3–27.8)% to 31.2 (17.3–39.1)% (*p* = 0.05) and by 21.4 [+7.0; +51.3]% from 38.4 (16.0–42.6)% to 49.2 (30.8–73.0)% (*p* = 0.04), respectively, while after vaccination with one dose the decrease in these parameters was 25.1 [45.6; +0.2]% from 53.1 (22.8–75.6)% to 7.3 (2.7–44.0)% (*p* < 0.001 for TLR3) and by 23.3 [57.6; 7.8]% from 66.1 (35.3–80.8)% to 6.3 (1.6–73.0)% (*p* = 0.01 for TLR9) (Figure 11a,b). The rate of change in the proportion of monocytes expressing TLR3 and TLR9 as a result of vaccination between the one- and two-dose groups differed statistically significantly (*p* = 0.01 in both cases) (Figure 12a,b).

As a result of vaccination with both one and two doses, the proportion of monocytes expressing TLR8 decreased statistically significantly—by 34.9 [67.0; 15.4]% from 67.5 (13.9–87.4)% to 7.4 (4.6–24.7)% (*p* < 0.001) in the single-dose group and by 39.0 [58.3; 10.3]% from 59.4 (26.2–62.2)% to 7.2 (1.5–23.1)% in the two-dose group (Figure 12c).

The intensity of the reduction (delta of the percentage) was not statistically significantly different between the one-dose and two-dose groups (*p* = 0.96).

To summarize, after vaccination with one dose there was a decrease in most indicators (the proportion of granulocytes expressing TLR3 and TLR9 and the proportion of monocytes expressing TLR3, TLR8, and TLR9), while after vaccination with two doses, on the contrary, an increase in some indicators was observed (the proportion of lymphocytes expressing TLR9, and the proportion of monocytes expressing TLR3 and TLR9).

### 3.3. Comparing the Content of TLR3, TLR8, and TLR9 in Healthy Volunteers after Vaccination with One Dose of the Quadrivalent Vaccine

On the granulocytes there were no dynamics in the level of TLR expression: TLR3 was 91.65 (88.13–96.63)% and after vaccination it became 93.35 (90.7–95.53)% (*p* = 0.57); TLR9 was 92.7 (87.3–95.3)% and after vaccination it became 91.55 (88.65–95.3)% (*p* = 0.67); and TLR8 was 86.5 (79.9–92.4)% before and became 85.03 (80.3–86.5)% (*p* = 0.27) after vaccination (Figure 13).

On lymphocytes there was a statistically significant increase: TLR3—from 0.10 (0–0.2)% up to 0.15 (0.1–0.225)% (*p* = 0.03), TLR8—from 0.1 (0–0.2)% up to 0.2 (0.1–0.3)% (*p* = 0.05), and TLR9—from 0.1 (0–0.1)% up to 0.15 (0.1–0.3)% (*p* = 0.02) (Figure 14).

On monocytes there is also a statistically significant increase: TLR3—from 8.6 (4.7–23.7)% to 23.6 (12.5–32.15)% (*p* = 0.03), TLR8—from 6.2 (2.8–17)% up to 15.25 (8.3–25.3)% (*p* = 0.05), and TLR9—from 12.45 (4.15–24.98)% up to 19.25 (9.7–43.98)% (*p* = 0.04) (Figure 15).

## 4. Discussion

Immunodeficiencies affecting humoral immunity represent a heterogeneous group of diseases, of which CVID is the most common. Common variable immune deficiency (CVID) is a primary immunodeficiency, a rare disease, diverse in clinical manifestations (phenotypes), and characterized by hypogammaglobulinemia [9]. Despite the fact that variants of monogenic forms of CVID have already been described, the mode of inheritance is mainly polygenic [9,39]. Clinical manifestations of CVID include primarily recurrent sinopulmonary infections, but patients also have an increased susceptibility to developing cancer, autoimmune, or inflammatory diseases. Autoimmune disorders are diagnosed in approximately 20–25% of patients with CVID [40].

The main laboratory sign of CVID is the inability of B cells to produce antibodies, and the main therapy is immunoglobulin replacement therapy to fight infections. Decades ago, Hermans et al. [41] were the first to describe the absence of vaccine-specific antibody synthesis in a cohort of patients with CVID. Subsequently, inadequate response to vaccination became one of the key diagnostic criteria.

At the moment, it is not completely known whether vaccination with inactivated vaccines against viral infections can have a sufficient preventive effect in patients with CVID, since large cohort studies have not been conducted due to the rare occurrence of this pathology in the population, as well as the delay in diagnosis for decades [42].

However, data obtained in recent years have led to a revision of the position on vaccination of patients with IEI. Currently, vaccination against respiratory infections is recommended for all patients by the same scheme as in the healthy population [7]. There is currently no reliable data on whether the immune response in patients with CVID will be sufficient (protective), and therefore the study of the feasibility of vaccination specifically against viral infections becomes even more relevant.

Our study compared the effectiveness of influenza vaccination in patients with CVID with one dose of quadrivalent adjuvanted and two doses of trivalent adjuvanted influenza vaccine without IVIG immunotherapy for 7 weeks. An immune response was expected due to the presence of an adjuvant in the vaccines that activates innate immune factors, as well as the expanded spectrum of antigens in the vaccine, in contrast to the study using the monovaccine Pandemrix [43].

Toll-like receptors (TLRs) play an important role in the formation of the post-vaccination immune response. TLRs are pattern recognition receptors (PRRs) that play a key role in the elimination of microbial agents. They initiate a series of signal cascades, constituting the primary line of defense against microorganisms through the recruitment of phagocytes or activation of dendritic cells [13,14,44]. Additionally, these signals trigger the maturation of dendritic cells, thereby orchestrating the secondary immune response known as acquired immunity. These are conserved membrane proteins that provide alternative modes of B cell activation. TLRs are expressed in a variety of cell types, including monocytes, phagocytic cells, dendritic cells, and B cell subsets. TLRs have established roles in the physiological regulation of pro-inflammatory cytokine production, essential for mounting immune responses against bacterial, fungal, and viral infections [45]. For example, TLR3 plays a pivotal role in cross-priming naive CD8 T cells, which subsequently differentiate into the cytotoxic T cells crucial for eliminating virus-infected cells [46,47], and the stimulation of TLR2, TLR4, and TLR9 results in respiratory burst and altered expression of adhesion molecules [48,49].

Dendritic cell activation is primarily associated with TLR2, TLR3, TLR4, TLR7, and TLR9. TLRs significantly contribute to the activation of antigen-presenting cells, not only by inducing pro-inflammatory cytokine production but also by enhancing the expression of various co-stimulatory molecules necessary for effective antibody recognition [50,51]. Furthermore, TLRs govern the dendritic cell maturation and antigen-presenting function [52] that is very important and plays a crucial role in susceptibility of patients with CVID toward infections. Studies have reported that influenza vaccines activate innate effectors—comprising both myeloid and lymphoid lineages of dendritic cells, which constitute the first line of defense against infections [53].

Besides its beneficial effects on a host’s innate immune responses, the influenza vaccine also includes enhancement of phagocytosis and its anti-toxic effects, such as the reduction in free radicals [54].

In 2022 there was an analysis conducted of the safety and immunogenicity profile of an azoximer bromide polymer-adjuvanted subunit influenza vaccine, which included trials performed between 1993 and 2016 and comprised 11,736 participants aged between 6 months and 99 years [55]. It showed that Grippol family vaccines induced antibody production in both children and adults up to 60 years at levels similar to vaccines with the standard amount of HA. In another study of the influence of adjuvanted vaccines with azoximer bromide on the effectors of inborn immunity it was shown that all evaluated (split, subunit, and adjuvanted) influenza vaccines elicited a statistically significant (*p* < 0.05) increase in the counts of granulocytes expressing TLR2, TLR6, TLR8, and TLR9 in peripheral blood mononuclear cell (PBMC) cultures when compared to unstimulated cells [56]. But unlike “classic” vaccines, adjuvanted vaccine showed high induction potential on TLR9- and TLR8-expressing cells, compared to the subunit vaccine (*p* = 0.012 and *p* < 0.001, respectively) and split vaccine (*p* = 0.003 and *p* < 0.001, respectively) possibly because of the co-stimulating effect of the adjuvant in the adjuvanted vaccine.

We studied expressions of TLR3, TLR8, and TLR 9, which like TLR7, are mainly located in the endosomal compartment; this location allows these receptors to recognize DNA and RNA breakdown products of viral and bacterial origin. TLR9 is expressed on human B cells. Activation of TLR3 on peripheral blood mononuclear cells, which include lymphocytes (T, B, and NK cells), monocytes and dendritic cells, as well as fibroblasts in CVID, leads to normal production of IFN-α and IFN-β, potentially providing adequate protection against viruses.

Our study also obtained data on the activation of TLR3 expressed on granulocytes and lymphocytes after repeated administration of one dose of adjuvanted influenza vaccine (as is recommended according to present guidelines—annually) and on monocytes upon administration of two doses simultaneously and, conversely, the absence of TLR3 activation upon the first vaccination with one dose. Consequently, annual vaccination against influenza using the same adjuvanted vaccine contributes to temporary nonspecific prevention of respiratory infections [57,58,59,60,61,62,63]. Simultaneous administration of two doses of vaccines also contributes to TLR3 expression which is observed only on monocytes, and, probably, the protective effect against other respiratory infections may be less pronounced.

One of the most powerful stimulators of B cell activation and maturation are endosomal toll-like receptors, the agonists of which are single-stranded RNA or various synthetic agonists (TLR7) and unmethylated CpG motifs in microbial DNA (CpG DNA) (TLR9), respectively [64]. Binding of TLR9 by CpG DNA has been shown to activate normal B cells, enhance the expression of co-stimulatory molecules, trigger the secretion of IL-6 and IL-10, and mediate T independent isotype switching and the production of specific antibodies independent of B cell receptor (BCR) binding [18,19,20,21,22]. Our study also found significant expression of TLR9, both one year after repeated vaccination with one dose and with the simultaneous administration of two doses of adjuvanted influenza vaccines, which was not observed after the first year of vaccination of patients with CVID with one dose. The role of TLR8, according to the results of our study, is less significant in the mechanisms of formation of post-vaccination immunity in patients with CVID, although one cannot fail to note the increase in its expression on lymphocytes a year after repeated vaccination with one dose of the study cohort of patients.

Moving on to the analysis of the data on TLR receptors in patients with CVID, it is necessary to indicate the results of studying similar indicators in healthy people, who were vaccinated as a control group and showed a statistically significant increase in the expression of all toll-like receptors, types 3, 8, and 9, on all studied immunocytes, except for granulocytes. While in patients with CVID there is an increase in the expression of intracellular receptors TLR3 and TLR9 also on granulocytes. On lymphocytes the dynamics of the expression of TLR3 and TLR8 is similar to that in healthy participants and is partially observed at the trend level after repeated vaccination a year later as well as with TLR9 after administration of two doses. On monocytes from patients with CVID, regardless of the immunization regimen, completely comparable results were obtained on the expression of all intracellular receptors, which probably indicates the activation of innate immune parameters and nonspecific protection against pathogens and an enhanced immune response.

## Figures and Tables

**Figure 1 viruses-15-02091-f001:**
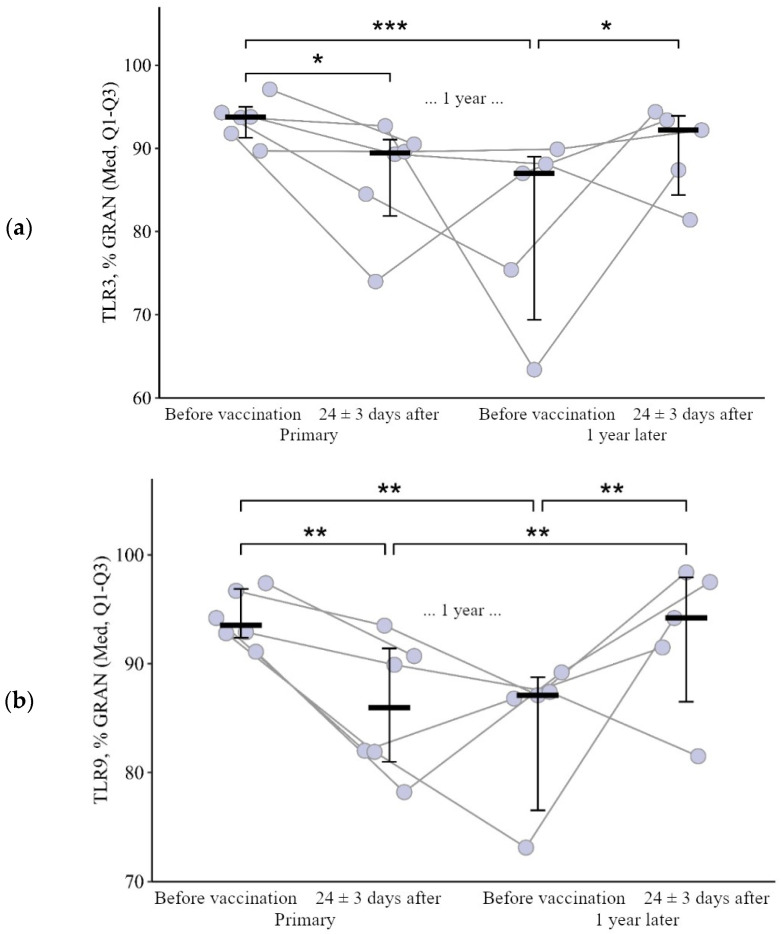
Percentage of granulocytes expressing TLR3 and TLR9 in patients with CVID at different stages of vaccination with a single vaccine dose (individual values as well as the median and interquartile range are given). *p*-value for comparing TLRs expression at different stages of vaccination (before vs. 24 ± 3 days after immunization with one vaccine dose during primary vs. a year later vaccination). * Statistically significant differences between different stages of vaccination at *p* ≤ 0.05 level. ** statistically significant differences between different stages of vaccination at *p* < 0.01 level. *** statistically significant differences between different stages of vaccination at *p* < 0.001. A robust linear mixed effects model was used for calculations.

**Figure 2 viruses-15-02091-f002:**
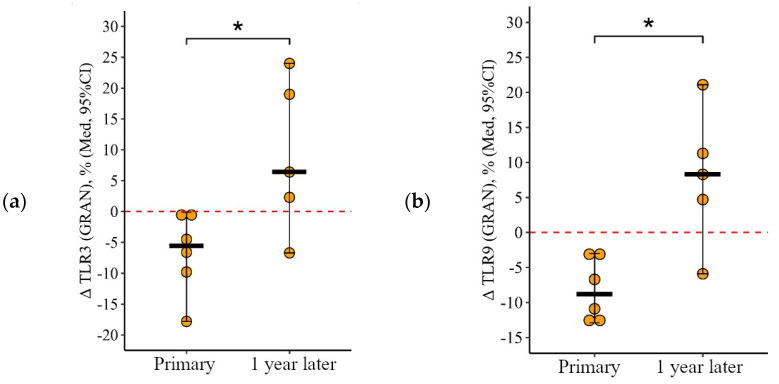
Delta (Δ—difference between the parameters before and 24 ± 3 days after vaccination) of the percentage of granulocytes expressing TLR3 and TLR9 in patients with CVID vaccinated with a single vaccine dose (individual values as well as the median and its 95% confidence interval are given). *p*-value for comparing delta of the percentage of granulocytes expressing TLR3 and TLR9 during primary and a year later vaccination. *—statistically significant differences between changes after primary and second vaccination at *p* ≤ 0.05 level. Mann–Whitney test was used.

**Figure 3 viruses-15-02091-f003:**
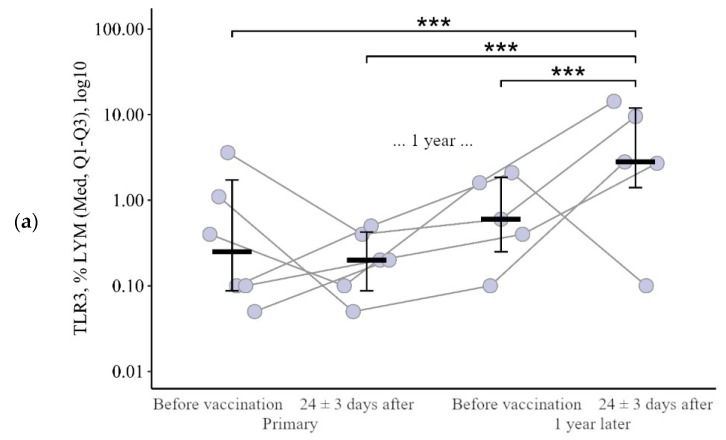
Percentage of lymphocytes expressing TLR3, TLR8, and TLR9 in patients with CVID at different stages of vaccination with a single vaccine dose (individual values as well as the median and interquartile range are shown). *p*-value for comparing TLRs expression at different stages of vaccination (before vs. 24 ± 3 days after immunization with one vaccine dose during primary vs. a year later vaccination).▪ Differences between different stages of vaccination at the *p* < 0.10 trend level. * Statistically significant differences between different stages of vaccination at *p* ≤ 0.05 level. ** Statistically significant differences between different stages of vaccination at *p* < 0.01 level. *** Statistically significant differences between different stages of vaccination at *p* < 0.001. A robust linear mixed effects model was used for calculations.

**Figure 4 viruses-15-02091-f004:**
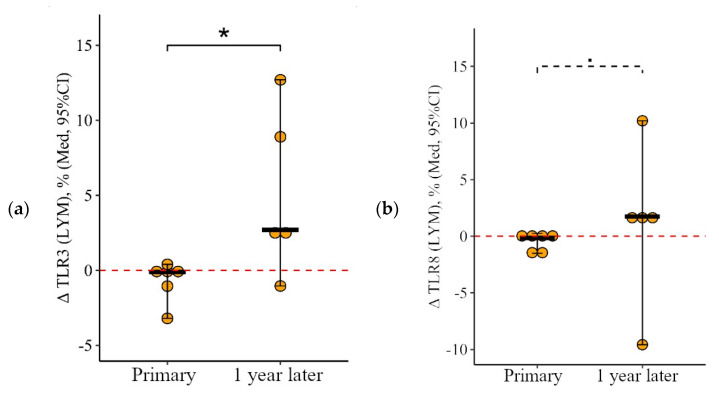
Delta (Δ—difference between the parameters before and 24 ± 3 days after vaccination) of the percentage of lymphocytes expressing TLR3 and TLR8 in patients with CVID vaccinated with a single vaccine dose, at various stages of vaccination (individual values are given, median and its 95% confidence interval). *p*-value for comparing delta of the percentage of lymphocytes expressing TLR3 and TLR8 during primary and a year later vaccination. ▪—differences between changes after
primary and second vaccination one year later at the *p* < 0.10 trend level. *—statistically significant differences between changes after primary and second vaccination one year later at *p* ≤ 0.05 level. Mann–Whitney test was used.

**Figure 5 viruses-15-02091-f005:**
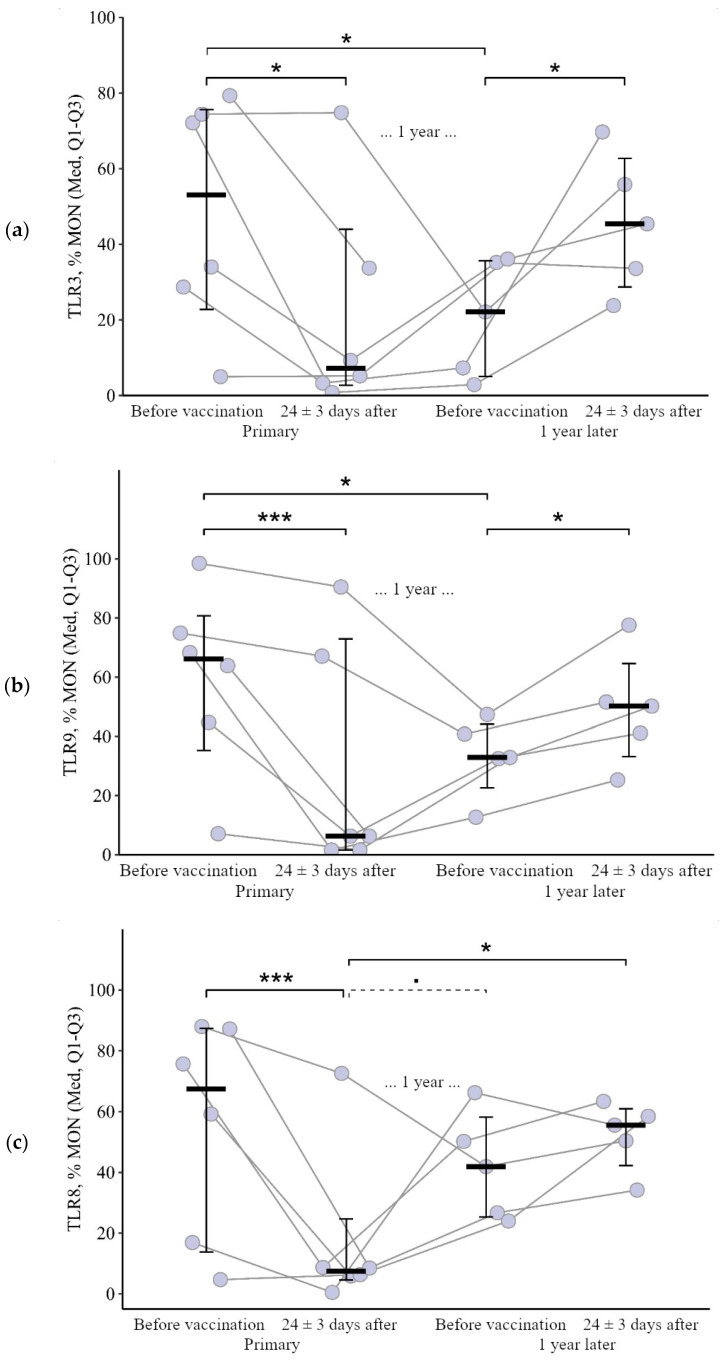
Percentage of monocytes expressing TLR3, TLR8, and TLR9 in patients with CVID at different stages of vaccination with a single vaccine dose (individual values as well as the median and interquartile range are shown). *P*-value for comparing TLRs expression at different stages of vaccination (before vs. 24 ± 3 days after immunization with one vaccine dose during primary vs. a year later vaccination). ▪ Differences between different stages of vaccination at the *p* < 0.10 trend level. * Statistically significant differences between different stages of vaccination at *p* ≤ 0.05 level. *** Statistically significant differences between different stages of vaccination at *p* < 0.001. A robust linear mixed effects model was used for calculations.

**Figure 6 viruses-15-02091-f006:**
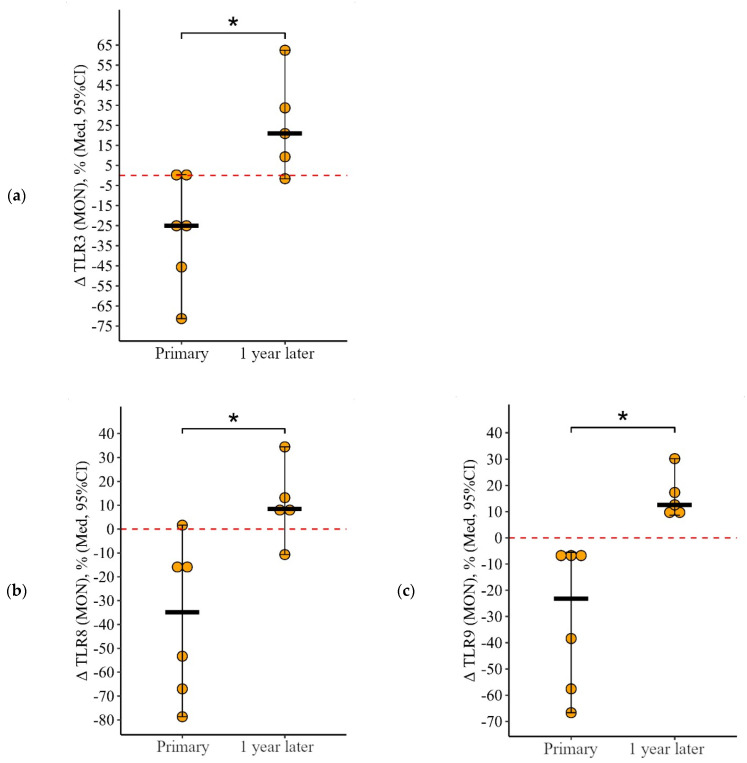
Delta (Δ—difference between the parameters before and 24 ± 3 days after vaccination) of the percentage of lymphocytes expressing TLR3, TLR8, and TLR9 in patients with CVID vaccinated with a single vaccine dose at various stages of vaccination (individual values are given as well as the median and its 95% confidence interval). *p*-value for comparing delta of the percentage of monocytes expressing TLR3, TLR8, and TLR9 during primary and a year later vaccination. ▪—differences between changes after primary and second vaccination one year later at the *p* < 0.10 trend level. *—statistically significant differences between changes after primary and second vaccination one year later at *p* ≤ 0.05 level. Mann–Whitney test was used.

**Figure 7 viruses-15-02091-f007:**
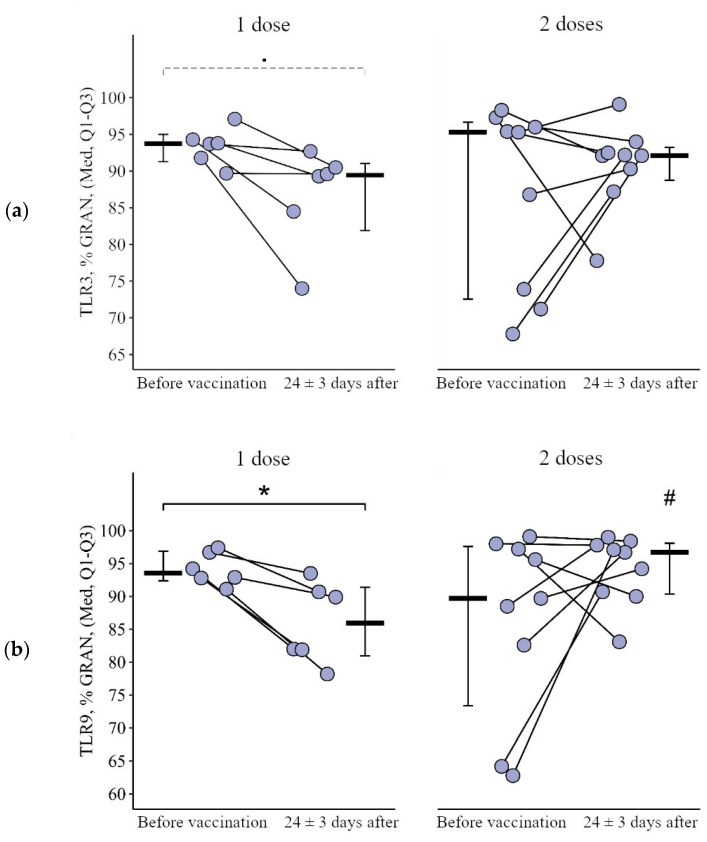
Percentage of granulocytes expressing TLR3 and TLR9 in patients with CVID after administration of single and double doses of adjuvanted influenza vaccine initially and 24 ± 3 days after immunization (individual values as well as the medians and interquartile range are given). *p*-value for comparing TLRs expression at different stages of vaccination (before vs. 24 ± 3 days after immunization with one vs. two vaccine doses. #—statistically significant differences between vaccination with one and two doses at the *p* ≤ 0.05 level. •—differences before and after vaccination at the *p* < 0.10 trend level. *—statistically significant differences before and after vaccination at *p* ≤ 0.05 level. A robust linear mixed effects model was used for calculations.

**Figure 8 viruses-15-02091-f008:**
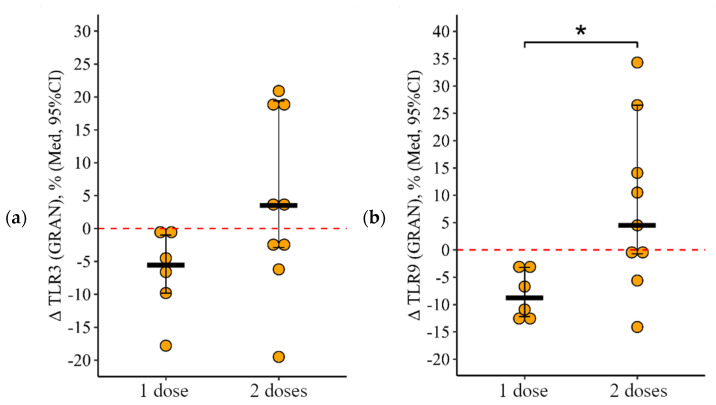
Delta (Δ—difference between the parameters before and 24 ± 3 days after vaccination) of the percentage of granulocytes expressing TLR3 and TLR9 in patients with CVID vaccinated with single and double doses (individual values as well as the median and its 95% confidence interval). *p*-value for comparing delta of the percentage of granulocytes expressing TLR3 and TLR9 after one vs. two vaccine doses administration. *—statistically significant differences between vaccination with one and two doses at the *p* ≤ 0.05 level. Mann–Whitney test was used.

**Figure 9 viruses-15-02091-f009:**
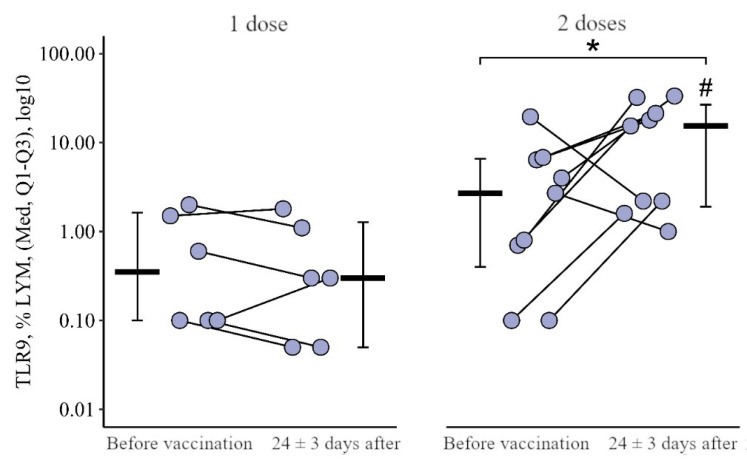
Percentage of lymphocytes expressing TLR9 in patients with CVID after administration of single and double doses of adjuvanted influenza vaccine initially and 24 ± 3 days after immunization (individual values as well as the medians and interquartile range are given). *—statistically significant differences before and after vaccination at *p* ≤ 0.05 level. #—statistically significant differences between vaccination with one and two doses at the *p* ≤ 0.05 level. A robust linear mixed effects model was used for calculations.

**Figure 10 viruses-15-02091-f010:**
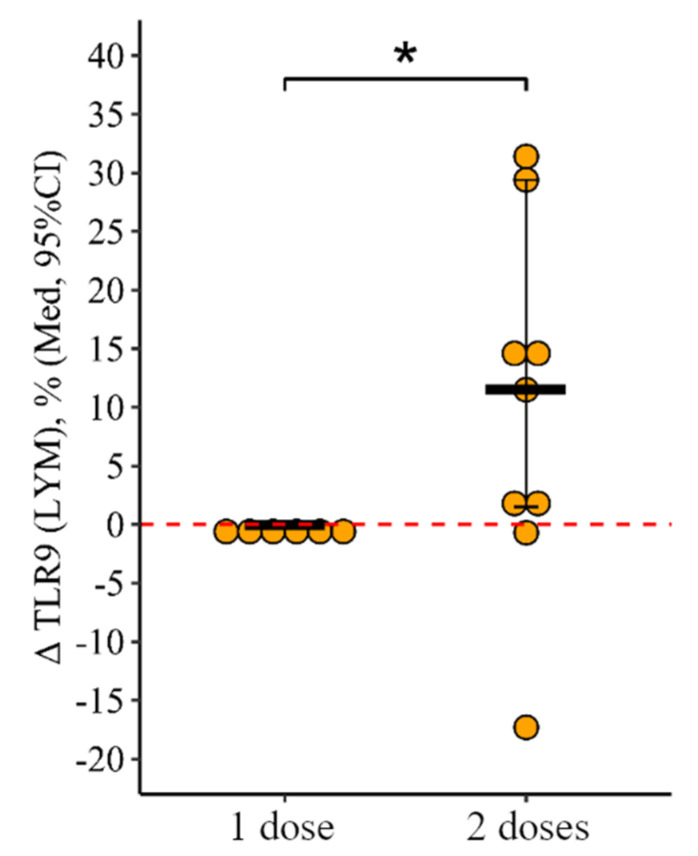
Delta (Δ—difference between the parameters before and 24 ± 3 days after vaccination) of the percentage of lymphocytes expressing TLR9 in patients with CVID vaccinated with single and double doses (individual values as well as the median and its 95% confidence interval). *—statistically significant differences between vaccination with one and two doses at the *p* ≤ 0.05 level. Mann–Whitney test was used.

**Figure 11 viruses-15-02091-f011:**
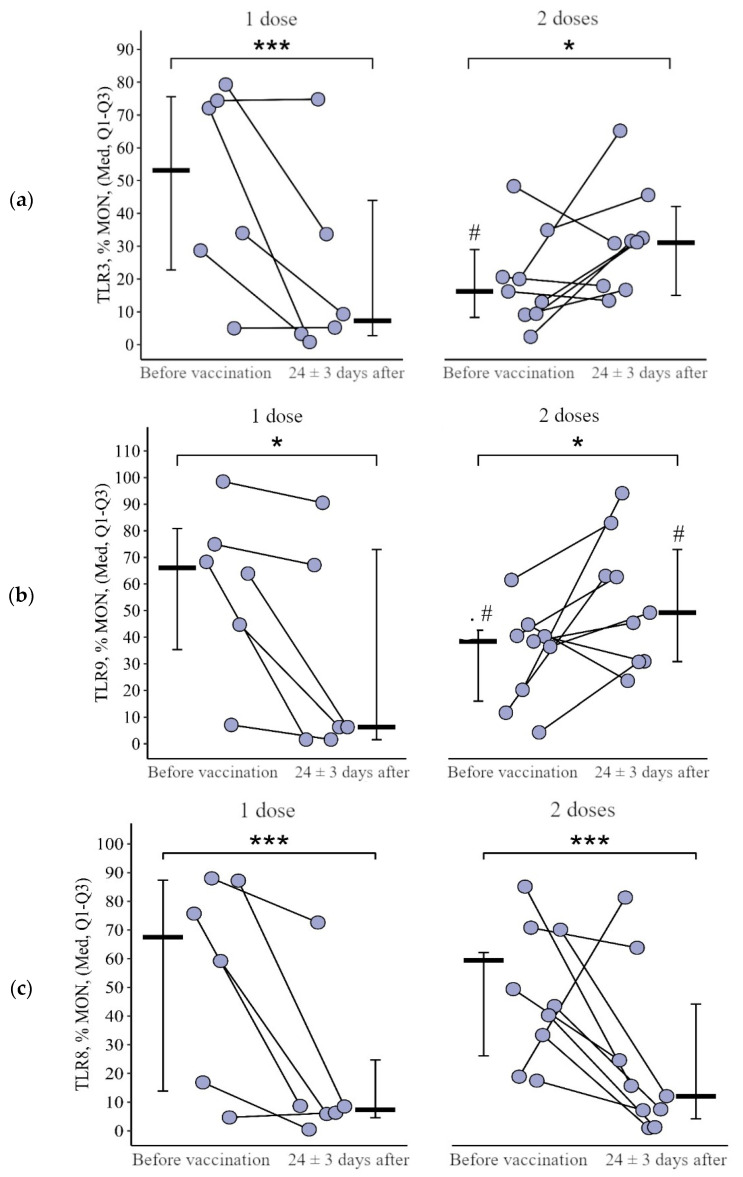
Percentage of monocytes expressing TLR3, TLR8, and TLR9 in patients with CVID after administration of single and double doses of adjuvanted influenza vaccine initially and 24 ± 3 days after immunization (individual values as well as the medians and interquartile range are given). *p*-value for comparing TLRs expression at different stages of vaccination (before vs. 24 ± 3 days after immunization with one vs. two vaccine doses). #—statistically significant differences between vaccination with one and two doses at the *p* ≤ 0.05 level. *—statistically significant differences before and after vaccination at *p* ≤ 0.05 level. ***—statistically significant differences before and after vaccination at *p* < 0.001 level. A robust linear mixed effects model was used for calculations.

**Figure 12 viruses-15-02091-f012:**
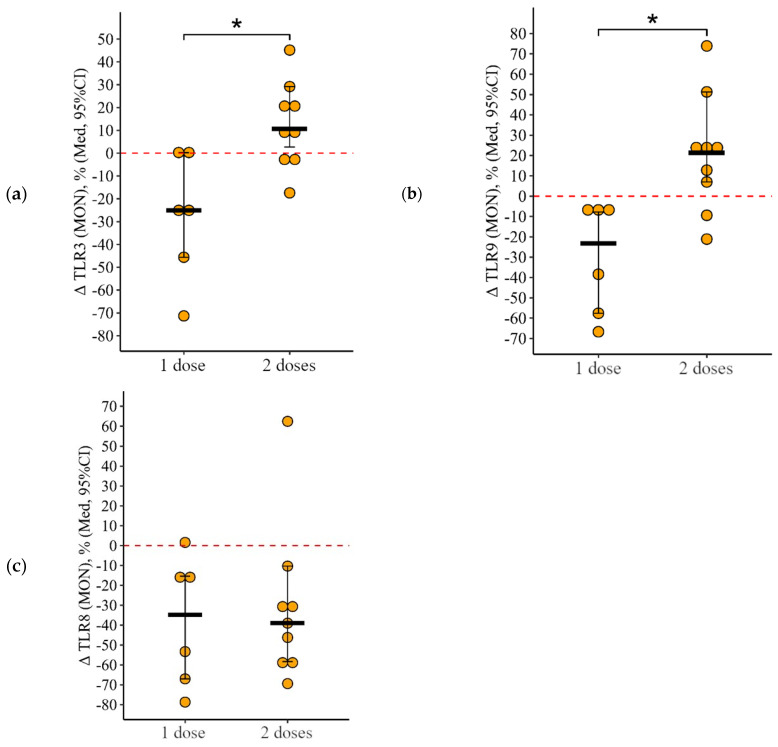
Delta (Δ—difference between the parameters before and 24 ± 3 days after vaccination) of the percentage of monocytes expressing TLR3, TLR8, and TLR9 in patients with CVID vaccinated with single and double doses (individual values as well as the median and its 95% confidence interval). *P*-value for comparing delta of the percentage of monocytes expressing TLR3, TLR8, and TLR9 after one vs. two vaccine doses administration. *—statistically significant differences between vaccination with one and two doses at the *p* ≤ 0.05 level. Mann–Whitney test was used.

**Figure 13 viruses-15-02091-f013:**
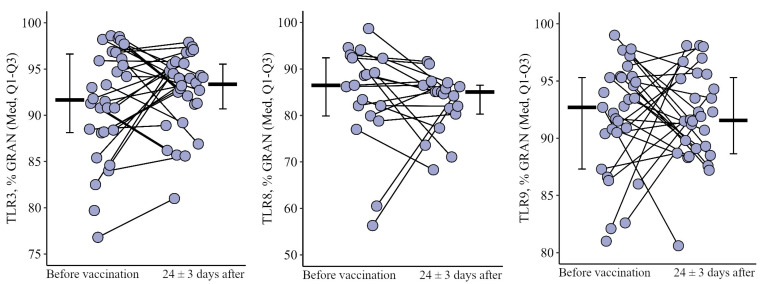
Percentage of granulocytes expressing TLR3, TLR8, and TLR9 in healthy volunteers after vaccination with one dose of the quadrivalent vaccine (individual values as well as the median and interquartile range are shown). Wilcoxon test for related samples was used.

**Figure 14 viruses-15-02091-f014:**
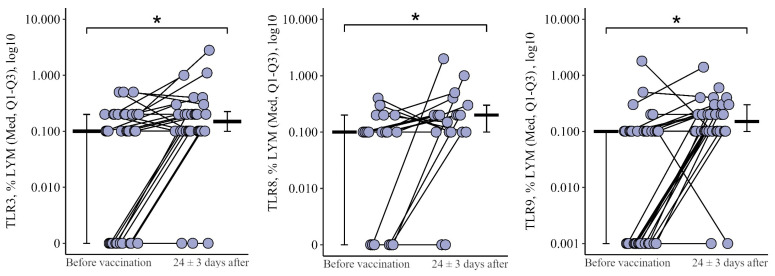
Percentage of lymphocytes expressing TLR3, TLR8, and TLR9 in healthy volunteers after vaccination with one dose of the quadrivalent vaccine (individual values as well as the median and interquartile range are shown). *—*p* ≤ 0.05. Wilcoxon test for related samples was used.

**Figure 15 viruses-15-02091-f015:**
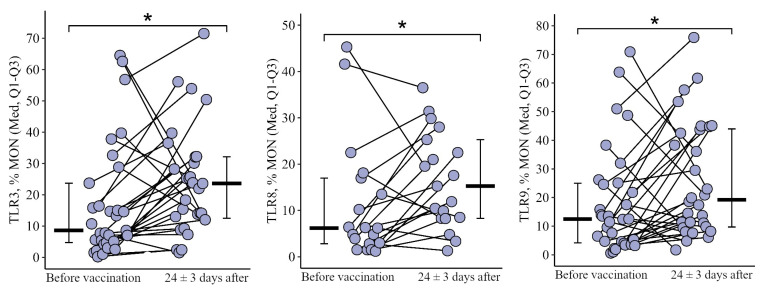
Percentage of monocytes expressing TLR3, TLR8, and TLR9 in healthy volunteers after vaccination with one dose of the quadrivalent vaccine (individual values as well as the median and interquartile range are shown). *—*p* ≤ 0.05. Wilcoxon test for related samples was used.

## Data Availability

Not applicable.

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
