# Peer review of "Expression of Toll-like Receptors on the Immune Cells in Patients with Common Variable Immune Deficiency after Different Schemes of Influenza Vaccination"

_viruses, 2023, doi:10.3390/v15102091_

Round 1

Reviewer 1 Report

General comments:

1) In this study, the individuals were vaccinated with aQIV and aTIV comprising purified hemagglutinin and neuraminidase obtained from influenza viruses A and B propagated in hen embryos. These vaccines are expected to have low/negligible levels of viral RNA. Furthermore, TLR3, 8 and 9 mainly recognise nucleic acids from pathogens (dsRNA, GU-rich ssRNA and unmethylated CpG DNA, respectively). Consequently these receptors are unlikely to be strongly triggered by the said vaccines. What immune responses are known to be activated by the adjuvant azoximer bromide? Is there evidence that TLRs, in particular, TLR3, 8, 9 are activated? How does immune response activated by azoximer bromide compare to with other adjuvants like MF59 and AS03? These topics should be elaborated in the introduction and discussion. 

2) Dosing regimens between control and most of the study groups are not very comparable.  Control group received single dose of aQIV (2018-9 season), whilst CIVD group 1a (1 individual) received single dose aQIV (2018-9), CVID group 1b (5 individuals) received 2 annual doses, single dose aQIV (2018-9) & single dose aTIV (2019-20), and CVID group 2 (9 individuals) received double dose aTIV (2019-20). How do the TLR expression data of immune cells in CVID individuals given one dose of aQIV (2018-9) compare against the data from control group also given the same? The small patient sample sizes used in the study is a weakness of the report.

Specific comments:

1) Abstract, line 13: not clear what the authors meant with the use of “perspective”. Can they rephrase for clarity?

2) Materials and Methods: the surface biomarkers used to identify the granulocytes, monocytes and lymphocytes for the flow cytometry studies should be stated.

3) Materials and Methods, Section 2.1: What is known about the levels of anti-IAV or IBV antibodies after the different vaccination regimens in the control and CVID individuals? This should be mentioned in the text.

4) Materials and Methods, Section 2.1: How any of the individuals in the control and CVID groups came down with influenza virus infection within the 24 days post-vaccination in the 2018-9 and 2019-20 seasons? This information should be provided in the text. How does this impact the data analysis and interpretation?

5) Did any of the individuals have prior influenza vaccinations before the 2018-9 aQIV and/or 2019-20 aTIV vaccination? This information should be provided.

6) What was the rationale for the double dose administered to CVID group 2?

7) Page 6, Section 2.1, 2nd last paragraph, lines 4-9: the “random factors” needs to be clarified, the brackets in the sentence look to be wrongly placed.

8) Page 6, Section 3.1.1 and other sections: given that the two doses were given one year apart and they target differing IAV strains, they should not be termed “primary” and “secondary” vaccinations. As such changes in TLRs in the different immune cells after aQIV and aTIV vaccinations should be interpreted independently.

9) Section 3.1.2: please clarify if only B lymphocytes were analysed.

10) Section 3.2: Comparison of these two CVID groups is not meaningful as the vaccines given to the two groups are not only different (aQIV vs aTIV) but also administered during different flu seasons (2018-9 vs 2019-20). 

Quality of English is fine.

Author Response

We are sincerely grateful to you for your valuable remarks and comments, your expert opinion and assessment!!!

Reviewer 2 Report

1. In my opinion, your paragraph introducing B-cells in the introduction section, starts "While B cells are traditionally" would be best served moving below the paragraph directly under it where you introduce a few specific TLRs. 

2. I recommend splitting the methods section up into subsections of specific experiments/methods to make it easier for readers to identify different methodologies. This is relatively standard for this journal. 

3. Please change figure 1, 3, 5, and 6 x-axis labels so that Primary and Secondary are grouped with 24 +/- 3days. This will make it easier to read/interpret.

4. In the last paragraph of 3.1.1 where you discuss TLR8 data, please specify "data not shown" or add in the graph. In my opinion, it is worth adding the graph to both Figure 1 & 2 despite no changes. 

5. Please combine figure 5 & 3. Also, add a panel in figure 4 to represent the TLR9 data. 

6. There is no mention of TLR7 in any of the experiments/text in section 3.1.1. Either amend the section title, or add data. 

17. 3.2.2 granulocytes section please state TLR8 data is not shown or add it (again, I think adding would be best).

8. 3.2.2 lymphocytes section please state TLR8/3 data is not shown or add it (again, I think adding would be best).

9. Please combine Figures 12 & 14, and add a panel for TLR8 in Figure 13

10. There is no mention of TLR7 anywhere in the data or physical results section yet it is stated in the materials and methods and section titles. This seems like a gross oversight. 

11. Minimally a supplemental figure needs to be added showing the data from the healthy control group. I would also like to see additional text in the results section at least describing and referencing this data. 

12. Personally I would like some additional experiments on these cells aside from just TLR content. Perhaps some functional assays to validate that your findings are meaningfully impacting immune cell outputs. 

1. Typo at the end of abstract, please change "researches of influence on" to "research on the influence of"

2. Typo in introduction of PAMPs and PRRs. Please add an s at the end of both acronyms as it is plural. 

3. Typo in next line, please change "signal" to signaling so it reads "endocytic and signaling"

Additional typos throughout, these are just some examples. 

Author Response

We are sincerely grateful to you for the valuable remarks and comments, your expert opinion and assessment!!!

Round 2

Reviewer 1 Report

The authors have adequately addressed this reviewer's queries. Minor correction for page 7, 1st line required, change sentence to "No one in the control nor CVID groups....".

Adequate. 

Reviewer 2 Report

I have no further comments, authors addressed my issues in a satisfactory manor.